# From Predictions to Decisions:
# Using L👀kahead Regularization

**Nir Rosenfeld**
Faculty of Computer Science
Technion - Israel Institute of Technology
nirr@cs.technion.ac.il

**Sophie Hilgard**
School of Engineering and Applied Science
Harvard University
ash798@g.harvard.edu

**Sai S. Ravindranath**
School of Engineering and Applied Science
Harvard University
saisr@g.harvard.edu

**David C. Parkes**
School of Engineering and Applied Science
Harvard University
parkes@eecs.harvard.edu

## Abstract

Machine learning is a powerful tool for predicting human-related outcomes, from creditworthiness to heart attack risks. But when deployed transparently, learned models also affect how users act in order to improve outcomes. The standard approach to learning predictive models is agnostic to induced user actions and provides no guarantees as to the effect of actions. We provide a framework for learning predictors that are accurate, while also considering interactions between the learned model and user decisions. For this, we introduce *look-ahead regularization* which, by anticipating user actions, encourages predictive models to also induce actions that improve outcomes. This regularization carefully tailors the uncertainty estimates that govern confidence in this improvement to the distribution of model-induced actions. We report the results of experiments on real and synthetic data that show the effectiveness of this approach.

## 1 Introduction

Machine learning is increasingly being used in domains that have considerable impact on people, ranging from healthcare [7], to banking [45], to manufacturing [52]. In many of these domains, fairness and safety concerns promote the deployment of fully transparent predictive models. An unavoidable consequence of this transparency is that end-users are prone to use models *prescriptively*: if a user (wrongly) views a predictive model as a description of the real world phenomena it models (e.g., heart attack risk), then she may look to the model for how to adapt her features in order to improve future outcomes (e.g., reduce her risk). But predictive models optimized for accuracy cannot in general be assumed to faithfully reflect post-modification outcomes, and model-guided actions can prove to be detrimental. Our goal in this paper is to present a learning framework for organizations seeking to deploy learned models in a way that is transparent *and* responsible, a setting we believe applies widely. Consider a medical center who would like to publish an online tool to allow users to estimate their heart attack risk, while keeping in mind that users may also infer how lifestyle changes will affect future risk.[1] Consider a lender who would like to be transparent about their first-time mortgage approval process, while knowing that this will suggest to applicants how they may alter their credit profiles. Consider a wine reseller, who would like to provide demand guidance

to producers through an interpretable model while considering that producers may use the same guidance to modify future vintages. Each of these organizations must be cognizant of the actions their classifiers promote, and seek to emphasize features that promote safe adaptations as well as predictive accuracy.

It is well understood that correlation and causation need not go hand-in-hand [32, 39]. What is novel about this work is that we seek models that serve the dual purpose of achieving predictive accuracy and providing high confidence that decisions made with respect to the model are safe. That is, we care foremost about the utility that comes from having a predictive tool, but recognize that these tools may also drive decisions.

To illustrate the potential pitfalls of a naïve predictive approach, consider a patient who seeks to understand his or her heart attack risk. If the patient consults a linear predictive model (as is often the case for medical models, see [51]), then a negative coefficient for alcohol consumption may lead the patient to infer that a daily glass of red wine would improve his or her prognosis. Is this decision justified? Perhaps not, although this recommendation has often been made based on correlative evidence and despite a clear lack of experimental support [18, 40].

Our main insight is that *controlling the tradeoff between accuracy and decision quality, where it exists, can be cast as a problem of model selection*. For instance, there may be multiple models with similar predictive performance but different coefficients, that therefore induce different decisions [5]. To achieve this tradeoff, we introduce *lookahead regularization*, which balances accuracy and the improvement associated with induced decisions. This is achieved by modeling how users will act, and penalizeing a model unless there is high confidence that decisions will improve outcomes.

Formally, these decisions induce a *target distribution* $p'$ on covariates that may differ from the distribution of data at training, $p$. In particular, a decision will map an individual with covariates $x$ to new covariates $x'$. For a prespecified confidence level $\tau$, we want to guarantee improvement for at least a $\tau$-fraction of the population, comparing outcomes under $p'$ in relation to outcomes in $p$ (under an invariance assumption on $p(y|x)$). The technical challenge is that $p'$ may differ considerably from $p$, resulting in uncertainty in estimating the effect of decisions. To solve this, lookahead regularization makes use of an uncertainty model that provides confidence intervals around decision outcomes. A discriminative uncertainty model is trained through importance weighting [14, 44, 48] to handle covariate shift, and is designed to estimate accurate intervals for $p'$.

Lookahead regularization has stages that alternate between optimizing the different components of our framework: the *predictive model* (under the lookahead regularization term), the *uncertainty model* (used within the regularization term), and the *propensity model* (used for covariate shift adjustment). If the uncertainty model is differentiable and the predictive model is twice-differentiable, then gradients can pass through the entire pipeline and gradient-based optimization can be applied. We run three experiments. One experiment uses synthetic data and illustrates the approach, helping to understand what is needed for it to succeed. The second experiment considers an application to wine quality prediction, and shows that even simple tasks lead to interesting tradeoffs between accuracy and improved decisions. The third experiment focuses on predicting diabetes progression and includes a demonstration of the framework in a setting with individualized actions.

## 1.1 Related work

**Strategic Classification.** In the field of *strategic classification*, the learner and agents engage in a Stackelberg game, where the learner attempts to publish a maximally accurate classifier taking into account that agents will shift their features to obtain better outcomes under the classifier [17]. While early efforts viewed all modifications as "gaming"— an adversarial effect to be mitigated [9, 6] —a recent trend has focused on creating incentives for modifications that lead to better outcomes *under the ground truth function* rather than simply better classifications [22, 2, 16, 49]. In the absence of a known mapping from effort to ground truth, Miller et al. [29] show that incentive design relates to causal modeling, and several responsive works explore how the actions induced by classifiers can facilitate discovery of these causal relationships [3, 43]. The second order effect of strategic classification on algorithmic fairness has also motivated several works [27, 20, 30]. Generally, these works consider the equilibrium effects of classifiers, where the choice of model affects covariate distributions and in turn predictive accuracy.

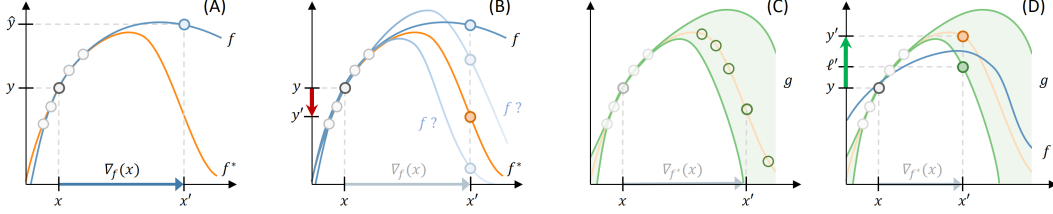

Figure 1: An illustration of our approach. Here $p(y|x)$ is deterministic, $y = f^*(x)$, and the data density $p(x)$ is concentrated to the left of the peak. (A) Users ($x$) seeking to improve their outcomes ($y$) often look to predictive models for guidance on how to act, e.g. by following gradient information ($x \mapsto x'$). (B) But actions may move $x'$ into regions of high uncertainty, where $f$ is unconstrained by the training data. Models of equally good fit on $p$ can behave very differently on $p'$, and hence induce very different decisions. (C) Promoting good decisions requires reasoning about the uncertainty in decision outcomes. For this, our approach learns an interval model $g(x') = [\ell', u']$ guaranteeing that $y' \in [\ell', u']$ with confidence $\tau$, decoupled from $f$ and targeted specifically at $p'$. (D) *Lookahead regularization* utilizes these intervals to balance between accuracy and improvement, achieved by penalizing $f$ whenever $y > \ell'$ (Eq. (4)). By incorporating into the objective a model of user behavior, our approach learns predictive models encouraging safe decisions, i.e., having $y' \geq y$ w.p. at least $\tau$.

Recent work on *performative prediction* [33] studies the equilibrium of retraining dynamics in settings where the model at each round affects the next input distributions (this generalizes strategic classification). Training is focused entirely on accuracy, and does not consider the quality of induced decision outcomes (these can be arbitrarily bad). We study a different setting of one-time interactions between users and a model (consider first-time mortgage buyers or consumers who access a medical risk calculator online), focusing on the tradeoff between predictive accuracy (as it relates to $p$) and decision outcomes (under $p'$). Our model is also relevant in settings with feedback coming in slowly, with models being intermittently re-trained, and where decision outcomes are consequential at each step of the retraining process.

**Causality, Covariate Shift, and Distributionally Robust Learning.** There are many efforts in ML to quantify the uncertainty associated with predictions and identify domain regions where models err [25, 19, 13, 15, 50, 26]. However, most methods fail to achieve desirable properties when deployed out of distribution (OOD) [47]. When the shifted distribution is unknown at train time, distributionally robust learning can provide worst-case guarantees for specific types of shifts but require unrealistic computational expense or restrictive assumptions on model classes [46]. Although we do not know ahead of training our shifted distribution of interest, our framework is concerned only with the single, specific OOD distribution that is induced by the learned predictive model. Hence, we need only guarantee robustness to this particular distribution, for which we make use of tools from learning under covariate shift [4]. Relevant to our task, Mueller et al. [31] seek to identify treatments which are beneficial with high probability under the covariate shift assumption. Because model variance generally increases when covariate shift acts on non-causal variables [34], our framework of trading off uncertainty minimization with predictive power relates to efforts in the causal literature to find models which have optimal predictive accuracy while being robust to classes of interventional perturbations [28, 38].

## 2 Method

Let $x \in \mathcal{X} = \mathbb{R}^d$ denote a feature vector and $y \in \mathbb{R}$ denote a label, where $x$ describes the object of interest (e.g., a patient, a customer, a wine vintage), and $y$ describes the quality of an outcome associated with $x$, where we assume that higher $y$ is better. We assume an observational dataset $\mathcal{S} = \{(x_i, y_i)\}_{i=1}^m$, which consists of IID samples from a population with joint distribution $(x, y) \sim p(x, y)$ over covariates (features) $x$ and outcomes $y$. We denote by $p(x)$ the marginal distribution on covariates.

Let $f : \mathcal{X} \to \mathbb{R}$ denote a model trained on $\mathcal{S}$. We assume that $f$ is used in two different ways:

1. **Prediction:** To predict outcomes $y$ for objects $x$, sampled from $p(x)$.

2. **Decision:** To take action, through changes to $x$, with the goal of improving outcomes.

We will assume that user actions map each $x$ to a new $x' \in \mathcal{X}$. We refer to $x'$ as a user's *decision* or *action* and denote decision outcomes by $y' \in \mathbb{R}$, We set $x' = d(x)$ and refer to $d : \mathcal{X} \to \mathcal{X}$ as the *decision function*. We will assume that users consult $f$ to drive decisions—either because they care only about predicted outcomes (e.g., the case of bank loans), or because they consider the model to be a valid proxy of the effect of a decision on the outcome (e.g., the case of heart attack risk or wine production). As in other works incorporating strategic users into learning [33, 17], our framework requires an explicit model of how users use the model to make decisions. For concreteness, we model users as making a step in the direction of the gradient of $f$, but note that the framework can also support any other differential decision model.[2] Since not all attributes may be susceptible to change (e.g., diet can be changed, height is fixed), we distinguish between *mutable* and *immutable* features using a task-specific *masking operator* $\Gamma : \mathcal{X} \to \{0, 1\}^d$.[3]

**Assumption 1** (User decision model). *Given masking operator $\Gamma$, we define user decisions as:*

$$x' = x + \eta \Gamma(\nabla_f(x)), \tag{1}$$

*where the* step size $\eta \geq 0$ *is a design parameter.*

Through Assumption 1, user decisions induce a particular decision function $d(x)$, and in turn, a *target distribution* over $\mathcal{X}$, which we denote $p'(x)$. This leads to a new joint distribution $(x', y') \sim p'(x, y)$, with decisions inducing new outcomes. To achieve causal validity in the way we reason about the effect of decisions on outcomes, we follow Peters et al. [34] and assume that $y$ depends only on $x$ and is invariant to the distribution on $x$.

**Assumption 2** (Covariate shift [44]). *The conditional distribution on outcomes, $p(y|x')$, is invariant for any arbitrary, marginal distribution $p'(x)$ on covariates, including the data distribution $p(x)$.*

Assumption 2 says that whatever the transform $d$, conditional distribution $p(y|x)$ is fixed, and the new joint distribution is $p'(x', y) = p(y|x')p'(x')$, for any $p'$ (note that $p'$ also depends on $f$). This covariate-shift assumption ensures the causal validity of our approach (and entails the property of *no-unobserved confounders*). Although a strong assumption, this kind of invariance is reasonable for many applications, and has been leveraged in other works that relate directly to questions of causality [36, 31], as well as more generally for settings in which the target and training distributions differ [41, 35, 44, 48].[4] There exist important domains in which violations are sufficiently minor that this is a reasonable assumption (see [31, 34] for discussion).

## 2.1 Learning objective

Our goals in designing a learning framework are twofold. First, we would like learning to result in a model whose predictions $\hat{y} = f(x)$ closely match the corresponding labels $y$ for $x \sim p(x)$. Second, we would like the model to induce decisions $x'$ for counterfactual distribution $p'$ whose outcome $y'$ improves upon the initial $y$. To balance between these two goals, we construct a learning objective in which a predictive loss function is augmented with a regularization term that promotes good decisions. The difficulty is that decision outcomes $y'$ depend on decisions $x'$ through the learned model $f$. Hence, realizations of $y'$ are unavailable at train time, as they cannot be observed until after the model is deployed. For this reason, simple constraints of the form $y' \geq y$ are ill-defined, and to regularize we must reason about outcome distributions $y' \sim p(y|x')$, for $x' \sim p'$. A naive approach might consider the average improvement, with $\mu' = \mathbb{E}_{y' \sim p(y|x')}[y']$, for a given $x' \sim p'$, and penalize the model whenever $\mu' < y$, for example linearly using $\mu' - y$. Concretely, $\mu'$ must be estimated, and since $f$ minimizes MSE, then $\hat{y}' = f(x')$ is a plausible estimate of $\mu'$, giving:

$$\min_{f \in F} \mathbb{E}_{p(x,y)}[(\hat{y} - y)^2] + \lambda \mathbb{E}_{p(x,y)}[\hat{y}' - y], \qquad \hat{y}' = f(x'), \tag{2}$$

where $\lambda \geq 0$ determines the relative importance of improvement over accuracy. Note that $x'$ depends on $f$ (see Eq. (1)).

There are two issues with this approach. First, learning can result in an $f$ that severely overfits in estimating $\mu$, meaning that at train time the penalty term in the (empirical) objective will appear to be low whereas at test time its (expected) value will be high. This can happen, for example, when $x'$ is moved to a low-density region of $p(x)$ where $f$ is unconstrained by the data and, if flexible enough, can artificially (and wrongly) signal improvement. To address this we use two decoupled models—one for predicting $y$ on distribution $p$, and another for handling $y'$ on distribution $p'$.

Second, in many applications it may be unsafe to guarantee that improvement hold only on average per individual (e.g., heart attack risk, credit scores). To address this, we encourage $f$ to improve outcomes with a certain degree of confidence $\tau$, for $\tau > 0$, i.e., such that $\mathbb{P}[y' \geq y] \geq \tau$ for a given $(x, y)$ and induced $x'$ and thus $p(y'|x')$. Importantly, while one source of uncertainty in $y'$ is $p(y|x')$, other sources of uncertainty exist, including those coming from insufficient data as well as model uncertainty. Our formulation is useful when additional sources of uncertainty are significant, such as when the model $f$ leads to actions that place $x'$ in low-density regions of $p$.

In our method, we replace the average-case penalty in Eq. (2) with a *confidence-based penalty*:

$$\min_{f \in F} \mathbb{E}_{p(x,y)}[(\hat{y} - y)^2] + \lambda \mathbb{E}_{p(x,y)}[\mathbb{1}\{\mathbb{P}[y' \geq y] < \tau\}], \qquad y' \sim p(y|x'), \qquad (3)$$

where $\mathbb{1}\{A\} = 1$ if $A$ is true, and 0 otherwise. In practice, $\mathbb{P}[y' \geq y]$ is unknown, and must be estimated. For this, we make use of an *uncertainty model*, $g_\tau : \mathcal{X} \to \mathbb{R}^2$, $g_\tau \in G$, which we learn, and maps points $x' \in \mathcal{X}$ to intervals $[\ell', u']$ that cover $y'$ with probability $\tau$. We also replace the penalty term in Eq. (3) with the slightly more conservative $\mathbb{1}\{\ell' < y\}$, and to make learning feasible we use the hinge loss $\max\{0, y - \ell'\}$ as a convex surrogate.[5] For a given uncertainty model, $g_\tau$, the empirical learning objective for model $f$ on sample set $\mathcal{S}$ is:

$$\min_{f \in F} \sum_{i=1}^{m} (\hat{y}_i - y_i)^2 + \lambda R(g_\tau; \mathcal{S}), \quad \text{where } R(g_\tau; \mathcal{S}) = \sum_{i=1}^{m} \max\{0, y_i - \ell'_i\}, \qquad (4)$$

where $R(g_\tau; \mathcal{S})$ is the *lookahead regularization* term.

By anticipating how users decide, this penalizes models whose induced decisions do not improve outcomes at a sufficient rate (see Figure 1).The novelty in the regularization term is that it accounts for uncertainty in assessing improvement, and does so for points $x'$ that are out of distribution. If $f$ pushes $x'$ towards regions of high uncertainty, then the interval $[\ell', u']$ is likely to be large, and $f$ must make more "effort" to guarantee improvement, something that may come at some cost to in-distribution prediction accuracy. As a byproduct, while the objective encodes the rate of decision improvement, we will also see the magnitude of improvement increase in our experiments.

Note that the regularization term $R$ depends both on $f$ and $g$—to determine $x'$, and to determine $\ell'$ given $x'$, respectively. This justifies the need for the decoupling of $f$ and $g$: without this, uncertainty estimates based on $f(x')$ are prone to overfit by artificially manipulating intervals to be higher than $y$, resulting in low penalization at train time without actual improvement (see Figure 2 (right)).

## 2.2 Estimating uncertainty

The usefulness of lookahead regularization relies on the ability of the uncertainty model $g$ to correctly capture the various kinds of uncertainties about the outcome value for the perturbed points. This can be difficult because uncertainty estimates are needed for out-of-distribution points $x'$.

Fortunately, for a given $f$ the counterfactual distribution $p'$ is known (by Assumption 1), and we can use the covariate transform associated with the decision to construct sample set $\mathcal{S}' = \{x'_i\}_{i=1}^{m}$. Even without labels for $\mathcal{S}'$, estimating $g$ is now a problem of *learning under covariate shift*, where the test distribution $p'$ can differ from the training distribution $p$. In particular, we are interested in learning uncertainty intervals that provide good coverage. There are many approaches to learning

under covariate shift. Here we describe the simple and popular method importance weighting, or inverse propensity weighting [44]. For a loss function $L(g) = L(y, g(x))$, we would like to minimize $\mathbb{E}_{p'(x,y)}[L]$. Let $w(x) = p'(x)/p(x)$, then by the covariate shift assumption:

$$\mathbb{E}_{p'(x,y)}[L(g)] = \int L(g)\,\mathrm{d}p'(x)\,\mathrm{d}p(y|x) = \int \frac{p'(x)}{p(x)}L(g)\,\mathrm{d}p(x)\,\mathrm{d}p(y|x) = \mathbb{E}_{p(x,y)}[w(x)L(g)].$$

Hence, training $g$ with points sampled from distribution $p$ but weighted by $w$ will result in an uncertainty model that is optimized for the counterfactual distribution $p'$. In practice, $w$ is itself unknown, but many methods exist for learning an approximate model $\hat{w}(x) \approx w(x)$ using sample sets $\mathcal{S}$ and $\mathcal{S}'$ (e.g. [21]). To remain within our discriminative approach, here we follow [4] and train a logistic regression model $h : \mathcal{X} \to [0, 1]$, $h \in H$, to differentiate between points $\tilde{x} \in \mathcal{S}$ (labeled $\tilde{y} = 0$) and $\tilde{x} \in \mathcal{S}'$ (labeled $\tilde{y} = 1$) and set weights to $\hat{w}(x) = e^{h(x)}$. As we are interested in training $g$ to gain a coverage guarantee, we define $L(y, g(x)) = \mathbb{1}\{y \notin [\ell, u]\}$ as in [37].

### 2.3 Algorithm

All the elements in our framework— the predictive model $f$, the uncertainty model $g$, and the propensity model $h$ —are interdependent. Specifically, optimizing $f$ in Eq. (4) requires intervals from $g$; learning $g$ requires weights from $h$; and $h$ is trained on $\mathcal{S}'$ which is in turn determined by $f$. Our algorithm therefore alternates between optimizing each of these components while keeping the others fixed. At round $t$, $f^{(t)}$ is optimized with intervals $[\ell'_i, u'_i] = g^{(t-1)}(x'_i)$, $g^{(t)}$ is trained using weights $w_i = h^{(t)}(x_i)$, and $h^{(t)}$ is trained using points $x'_i$ as determined by $f^{(t)}$. The procedure is initialized by training $f^{(0)}$ without the lookahead term $R$. For training $g$ and $h$, weights $w_i = \hat{w}(x_i)$ and points $\{x'_i\}_{i=1}^{m}$, respectively, can be precomputed and plugged into the objective. Training $f$ with Eq. (4), however, requires access to the *function* $g$, since during optimization, the lower bounds $\ell'$ must be evaluated for points $x'$ that vary as updates to $f$ are made. Hence, to optimize $f$ with gradient methods, we use an uncertainty model $g$ that is differentiable, so that gradients can pass through them (while keeping their parameters fixed). Furthermore, since gradients must also pass through $x'$ (which includes $\nabla_f$), we require that $f$ be twice-differentiable.

In the experiments we consider two methods for learning $g$:

1. Bootstrapping [11], where a collection of models $\{g^{(i)}\}_{i=1}^{k}$ is trained for prediction each on a subsampled dataset and combined to produce a single interval model $g$, and

2. Quantile regression [23], where models $g^{(\ell)}, g^{(u)}$ are discriminatively trained to estimate the $\tau$ and $1 - \tau$ quantiles, respectively, of the counterfactual target distribution $p'(y|x')$.

## 3 Experiments

In this section, we evaluate our approach in three experiments of increasing complexity and scale, where the first is synthetic and the latter two use real data. Because the goal of regularization is to balance accuracy with decision quality, we will be interested in understanding the attainable frontier of accuracy vs. improvement. For our method, this will mostly be controlled by varying lookahead regularization parameter, $\lambda \geq 0$. In all experiments we measure predictive accuracy with root mean squared error (RMSE), and decision quality in two ways: *mean improvement rate* $\mathbb{E}[\mathbb{1}\{y'_i > y_i\}]$ (corresponding to the the regularization term in Eq. (3)), and *mean improvement magnitude* $\mathbb{E}[y'_i - y_i]$ (corresponding to its convex proxy in Eq. (4)).[6]

To evaluate the approach, we need a means for evaluating counterfactual outcomes $y'$ for decisions $x'$. Therefore, and similarly to Shavit and Moses [42], we make use of an inferred 'ground-truth' function $f^*$ to test decision improvement, assuming $y' = f^*(x')$. Model $f^*$ is trained on the entirety of the data. By optimizing $f^*$ for RMSE, we think of this as estimating the conditional mean of $p(y|x)$, with the data labels as (arbitrarily) noisy observations. To make for an interesting experiment, we learn $f^*$ from a function class $F^*$ that is more expressive than $F$ or $G$. The sample set $\mathcal{S}$ will contain a small and possibly biased subsample of the data, which we call the 'active set', and that plays the role of a representative sample from $p$. This setup allows us not only to evaluate improvement, but also to experiment with the effects of different sample sets.

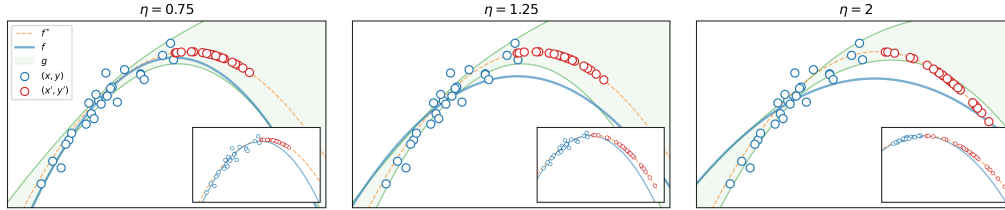

Figure 2: Results for the synthetic experiment comparing lookahead (main plots) to a baseline model (inlays; note the change in scale across plots). In both models, decisions move points $x$ over the peak. Under the baseline model, as $\eta$ increases, decision outcomes $y'$ worsen. Lookahead corrects for this, and at a small cost to accuracy (on $p$) ensures good decision outcomes (on $p'$, with sufficient overlap).

## 3.1 Experiment 1: Quadratic curves

For a simple setting, we explore the effects of regularized and unregularized learning on decision quality in a stylized setting using unidimensional quadratic curves. Let $f^*(x) = -x^2$, and assume $y = f(x) + \varepsilon$ where $\varepsilon$ is independently, normally distributed. By varying the decision model step-size $\eta$, we explore three conditions: one where a naïve approach works well, one where it fails but regularization helps, and one where regularization also fails.

In Figure 2 (left), $\eta$ is small, and the $x'$ points stay within the high certainty region of $p$. Here, the baseline works well, giving both a good fit and effective decisions, and the regularization term in the lookahead objective remains inactive. In Figure 2 (center), $\eta$ is larger. Here, the baseline model pushes $x'$ points to a region where $y'$ values are low. Meanwhile, the lookahead model, by incorporating into the objective the decision model and estimating uncertainty surrounding $y'$, is able to adjust the model to induce good decisions with some reduction in accuracy. In Figure 2 (right), $\eta$ is large. Here, the $x'$ points are pushed far into areas of high uncertainty. The success of lookahead relies on the successful construction of intervals at $p'$ through the successful estimation of $w$, and may fail if $p$ and $p'$ differ considerably, as is the case here.

## 3.2 Experiment 2: Wine quality

The second experiment focuses on wine quality using the wine dataset from the UCI data repository [10]. The wine in the data set has 13 features, most of which correlate linearly with quality $y$, but two of which (alcohol and malic acid) have a non-linear U-shaped or inverse-U shaped relationship with $y$. For the ground truth model, we set $f^*(x) = \sum_i \theta_i x_i + \sum_i \theta'_i x_i^2$ (RMSE = 0.2, $y \in [0,3]$) so that it captures these nonlinearities. To better demonstrate the capabilities of our framework, we sample points into the active set non-uniformly by thresholding on the non-linear features. The active set includes $\sim 30\%$ of the data, and is further split 75-25 into a train set used for learning and tuning and a held-out test set used for final evaluation.

For the predictive model, our focus here is on linear models. The baseline includes a linear $f_{\text{base}}$ trained with $\ell_2$ regularization (i.e., Ridge Regression) with regularization coefficient $\alpha \geq 0$. Our lookahead model includes a linear $f_{\text{look}}$ trained with lookahead regularization (Eq. (4)) with regularization coefficient $\lambda \geq 0$. In some cases we will add to the objective an additional $\ell_2$ term, so that for a fixed $\alpha$, setting $\lambda = 0$ recovers the baseline model. Lookahead was trained for 10 rounds and the baseline with a matching number of overall epochs. The uncertainty model $g$ uses residuals-based bootstrapping with 20 linear sub-models. The propensity model $h$ is also linear. We consider two settings: one where all features (i.e., wine attributes) are mutable and using decision step-size $\eta = 0.5$, and another where only a subset of the features are mutable and using step-size $\eta = 2$.

**Full mutability.** Figure 3 (left) presents the frontier of accuracy vs. improvement on the test set when all features are mutable. The baseline and lookahead models coincide when $\alpha = \lambda = 0$. For the baseline, as $\alpha$ increases, predictive performance (RMSE) displays a typical learning curve with accuracy improving until reaching an optimum at some intermediate value of $\alpha$. Improvement, however, monotonically decreases with $\alpha$, and is highest with no regularization ($\alpha = 0$). This is because in this setting, gradients of $f_{\text{base}}$ induce reasonably good decisions: $f_{\text{base}}$ is able to approximately recover the

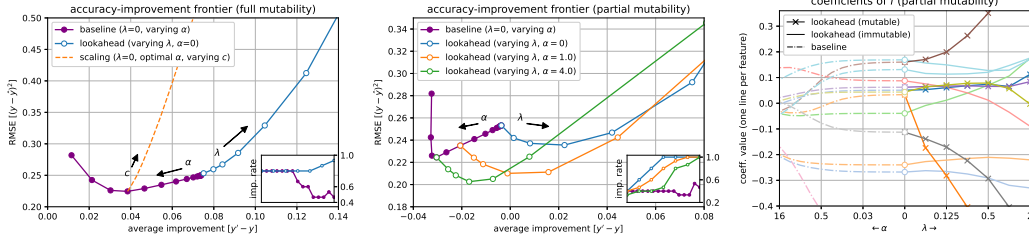

Figure 3: Results for the wine experiment. Tradeoff in accuracy and improvement under full mutability (left) and partial mutability (center), for which model coefficients are also shown (right).

dominant linear coefficients of $f^*$, and shrinkage due to higher $\ell_2$ penalization reduces the magnitude of the (typically positive, on average) change. With lookahead, increasing $\lambda$ leads to better decisions, but at the cost of higher (albeit sublinear) RMSE. The initial improvement rate at $\lambda = 0$ is high, but lookahead and $\ell_2$ penalties have opposing effects on the model. Here, improvement is achieved by (and likely requires) increasing the size of the coefficients of linear model, $f_{\text{look}}$. We see that $f_{\text{look}}$ learns to do this in an efficient way, as compared to a naïve scaling of the predictively-optimal $f_{\text{base}}$.

**Partial mutability.** Figure 3 (center) presents the frontier of accuracy vs. improvement when only a subset of the features are mutable (note that this effects the scale of possible improvement). The baseline presents a similar behavior to the fully-mutable setting, but with the optimal predictive model inducing a negative improvement. Here we consider lookahead with various degrees of additional $\ell_2$ regularization. When $\alpha = \lambda = 0$, the models again coincide. However, for larger $\lambda$, significant improvement can be gained with very little or no loss in RMSE, while moderate $\lambda$ values improve both decisions and accuracy. This holds for various values of $\alpha$, and setting $\alpha$ to the optimal value of $f_{\text{base}}$ results in lookahead dominating the trade-off curve for all observed $\lambda$. Improvement is reflected in magnitude and rate, which rises quickly from the baseline's $\sim 40\%$ to an optimal $100\%$, showing how lookahead learns models that lead to safe decisions.

Figure 3 (right) shows how the coefficients of $f_{\text{base}}$ and $f_{\text{look}}$ change as $\alpha$ and $\lambda$ increase, respectively (for lookahead $\alpha = 0$). As can be seen, lookahead works by making substantial changes to mutable coefficients, sometimes reversing their sign, with milder changes to immutable coefficients. Lookahead achieves improvement by capitalizing on its freedom to learn a useful direction of improvement within the mutable subspace, while compensating for the possible loss in accuracy through mild changes in the immutable subspace.

### 3.3 Experiment 3: Diabetes

The final experiment focuses on the prediction of diabetes progression using the diabetes dataset[7] [12]. The dataset has 10 features describing various patient attributes. We consider two features as mutable: BMI and T-cell count (marked as 's1'). While both display a similar (although reversed) linear relationship with $y$, feature s1 is much noisier. The setup is as in wine but with two differences: to capture nonlinearities we set $f^*$ to be a flexible generalized additive model (GAM) with splines of degree 10 (RMSE = 0.15), and train and test sets are sampled uniformly from the data. We normalize $y$ to $[0, 1]$ and set $\eta = 5$. Appendix C.4 includes a sensitivity analysis to learning with misspecified $\eta$.

Figure 4 (left) presents the accuracy-improvement frontier for linear $f$ and bootstrapped linear $g$. Results show a similar trend to the wine experiment, with lookahead providing improved outcomes (both rate and magnitude) while preserving predictive accuracy. Here, lookahead improves results by learning to increase the coefficient of s1, while adjusting other coefficients to maintain reasonable uncertainty. The baseline fails to utilize s1 for improvement since from a predictive perspective there is little value in placing weight on s1.

When $f$ is linear, decisions are uniform across the population in that $\nabla_{f_\theta}(x) = \theta$ is independent of $x$. To explore individualized actions, we also consider a setting where $f$ is a more flexible quadratic model (i.e., linear in $x$ and $x^2$) in which gradients depend on $x$ and uncertainty is estimated using quantile regression. Figure 4 (center) shows the data as projected onto the subspace $(x_{\text{BMI}}, x_{\text{s1}})$,

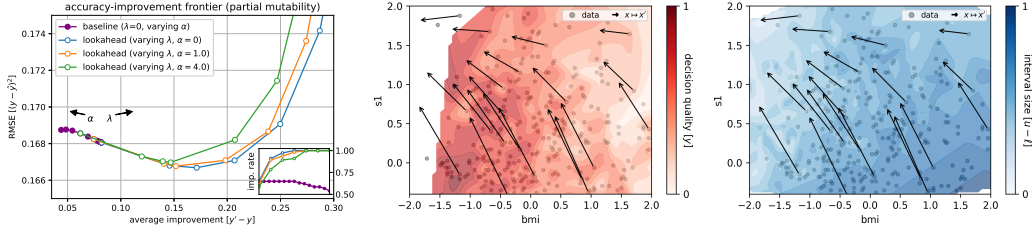

Figure 4: Results for the diabetes experiment. Tradeoff in accuracy and improvement under linear $f$ with partial mutability (left), visualization of shift $p \to p'$ with non-linear $f$ to regions of higher decision quality (center), and regions of lower uncertainty (right).

with color indicating outcome values $f^*(x)$, interpolated within this subspace. As can be seen, the mapping $x \mapsto x'$ due to $f_{\text{look}}$ generally improves outcomes. The plot reveals that, had we had knowledge of $f^*(x)$, uniformly decreasing BMI would also have improved outcomes, and this is in fact the strategy envoked by the linear $f_{\text{base}}$. But decisions must be made based on the sample set, and so uncertainty must be taken into account. Figure 4 (right) shows a similar plot but with color indicating uncertainty estimates as measured by the interval sizes given by $g$. The plot shows that decisions are directed towards regions of lower uncertainty (i.e., approximately following the negative gradients of the uncertainty slope), showing how lookahead successfully utilizes these uncertainties to adjust the predictive model $f_{\text{look}}$.

## 4    Discussion

Given the extensive use of machine learning across an ever-growing range of applications, we think it is appropriate to assume, as we have here, that predictive models will remain in widespread use, and that at the same time, and despite well-understood concerns, users will continue to act upon them. In line with this, our goal with this work has been to develop a machine learning framework that accounts for decision making by users but remains fully within the discriminative framing of statistical machine learning. The lookahead regularization framework that we have proposed augments existing machine learning methodologies with a component that promotes good human decisions. We have demonstrated the utility of this approach across three different experiments, one on synthetic data, one on predicting and deciding about wine, and one on predicting and deciding in regard to diabetes progression. We hope that this work will inspire continued research in the machine learning community that embraces predictive modeling while also being cognizant of the ways in which our models are used.

## Broader Impact

In our work, the learning objective was designed to align with and support the possible use of a predictive model to drive decisions by users. It is our belief that a responsible and transparent deployment of models with "lookahead-like" regularization components should avoid the kinds of mistakes that can be made when predictive methods are conflated with causally valid methods.

At the same time, we have made a strong simplifying assumption, that of covariate shift, which requires that the relationship between covariates and outcome variables is invariant as decisions are made and the feature distribution changes. This strong assumption is made to ensure validity for the lookahead regularization, since we need to be able to perform inference about counterfactual observations. As discussed by Mueller et al. [31] and Peters et al. [34], there exist real-world tasks that reasonably satisfy this assumption, and yet at the same time, other tasks— notably those with unobserved confounders —where this assumption would be violated. Moreover, this assumption is not testable on the observational data. This, along with the need to make an assumption about the user decision model, means that an application of the method proposed here should be done with care and will require some domain knowledge to understand whether or not the assumptions are plausible.

Furthermore, the validity of the interval estimates requires that any assumptions for the interval model used are satisfied and that weights $w$ provide a reasonable estimation of $p'/p$. In particular, fitting to

$p'$ which has little to no overlap with $p$ (see Figure 2) may result in underestimating the possibility of bad outcomes.

If used carefully and successfully, then the system provides safety and protects against the misuse of a model. If used in a domain for which the assumptions fail to hold then the framework could make things worse, by trading accuracy for an incorrect view of user decisions and the effect of these decisions on outcomes.

We would also caution against any specific interpretation of the application of the model to the wine and diabetes data sets. We note that model misspecification of $f^*$ could result in arbitrarily bad outcomes, and estimating $f^*$ in any high-stakes setting requires substantial domain knowledge and should err on the side of caution. We use the data sets for purely illustrative purposes because we believe the results are representative of the kinds of results that are available when the method is correctly applied to a domain of interest.

## Footnotes

[1]For example, the Mayo Clinic, a leading medical center in the U.S., provides such a calculator [8]. MDCalc.com is an example of a site that provides medical many risk assessment calculators to the public.

[2]Many works consider a 'rational' decision model $x' = \text{argmax}_{z \in \mathcal{X}} f(z) - c(x, z)$ where $c$ is a cost function. Eq. (1) can be thought of as modeling a boundedly-rational agent, taking action to optimize a local first-order approximation of $f$, with $\eta$ serving as a hard constraint. Our choice flows mostly for reasons of feasibility; in principal, rational decision models can be incorporated into learning using differentiable optimization layers (e.g., [1]). Like cost functions in all other works, $\eta$ is a design choice that can be set e.g. by an expert.

[3]$\Gamma$ does not reflect any causal assumptions; it merely states which features are amenable to change.

[4]One possibility to relax covariate shift is to instead assume Lipschitzness, i.e., that $p(y|x)$ changes smoothly with changes to $p'(x)$. This would affect the correctness of propensity weights, but can be accounted for by smoothly increasing uncertainty intervals (or reducing $\tau$). We leave this to future work.

[5]The penalty is conservative in that it considers only one-sided uncertainty, i.e., $y' < \ell$ and $u$ is not used explicitly. Although open intervals suffice here, most methods for interval prediction consider closed intervals, and in this way our objective can support them. For symmetric intervals, $\tau$ simply becomes $\tau/2$.

[6]Our code can be found at `https://github.com/papushado/lookahead`.

[7] https://www4.stat.ncsu.edu/~boos/var.select/diabetes.html

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
