[Supplementary Material · lookahead_supp.pdf]

# Appendix A  Pseudocode

Our algorithm alternates between optimizing the three components of the framework: a predictive model, a propensity model, and an uncertainty model. Here we give pseudocode for the following per-component objectives:

1. A predictive model $\hat{y} = f(x)$, optimizing the squared loss:

$$L_{\text{pred}}(f; \mathcal{S}) = \sum_{i=1}^{m} (y_i - \hat{y}_i)^2$$

2. A propensity weight model $w = e^{h(x)}$, optimizing the log-loss:

$$L_{\text{prop}}(h; \mathcal{S}, \mathcal{S}') = \sum_{i=1}^{m} \log(1 + e^{h(x_i)}) + \log(1 + e^{-h(x_i')})$$

3. An uncertainty interval model $[\ell, u] = g_\tau(x)$, optimizing the $\tau$-quantile loss:

$$L_{\text{uncert}}^{(\tau)}(g; \mathcal{S}, w) = \sum_{i=1}^{m} w(x_i) \max\{(\tau - 1)(y_i - \ell_i), \tau(y_i - \ell_i)\}$$

but note that others can be plugged in. The pseudocode is given below.

---

**Algorithm 1** Lookahead$(\mathcal{S}, T, \lambda, \eta, \tau)$

---

1: $f^{(0)} \leftarrow \text{argmin}_{f \in F} L_{\text{pred}}(f; \mathcal{S})$
2: **for** $t = 1, \ldots, T$ **do**
3:      $x_i' \leftarrow d_\eta(x_i; f^{(t-1)})$ for all $i = 1, \ldots, m$          $\triangleright$ e.g., $d_\eta(x; f) = x + \eta\Gamma(\nabla_f(x))$
4:      $\mathcal{S}' \leftarrow \{x_i'\}_{i=1}^{m}$
5:      $h^{(t)} \leftarrow \text{argmin}_{h \in H} L_{\text{prop}}(h; \mathcal{S}, \mathcal{S}')$
6:      $w \leftarrow e^{h^{(t)}}$
7:      $g^{(t)} \leftarrow \text{argmin}_{g \in G} L_{\text{uncert}}^{(\tau)}(g; \mathcal{S}, w)$
8:      $f^{(t)} \leftarrow \text{argmin}_{f \in F} L_{\text{pred}}(f; \mathcal{S}) + \lambda R(g^{(t)}; \mathcal{S})$
9: **return** $f^{(T)}$

---

# Appendix B  Uncertainty models

Here we describe the two uncertainty methods used in our paper and how they apply to our setting.

## B.1  Bootstrapping

Bootstrapping produces uncertainty intervals by combining the outputs of a collection of $k$ models $\{g^{(i)}\}_{i=1}^{k}$, each trained independently for *prediction* on a random subset of the data. There are many approaches to bootstrapping, and here we describe two:

- **Vanilla bootstrapping**: Each $g^{(i)}$ is trained using a predictive objective (e.g., squared loss) on a sample set $\mathcal{S}^{(i)} = \{(x_j^{(i)}, y_j^{(i)})\}_{j=1}^{m}$ where $(x_j^{(i)}, y_j^{(i)})$ are sampled with replacement from $\mathcal{S}$. The sub-models are then combined using:

$$g(x) = [\mu(x) - z\sigma(x), \mu(x) + z\sigma(x)]$$

where:

$$\mu(x) = \frac{1}{k} \sum_{i=1}^{k} g^{(i)}(x), \qquad \sigma(x) = \frac{1}{k} \sum_{i=1}^{k} (\mu(x) - g^{(i)}(x))^2$$

and $z$ is the z-score corresponding to the confidence parameter $\tau$ under a normal distribution.

- **Bootstrapping residuals**: First, a predictive model $\bar{g}$ is fit to the data, and residuals $r = y - \bar{g}(x)$ are computed. Then, each $g^{(i)}$ is trained on the original sample data but with ground truth-labels $y_i$ replaced with random pseudo-labels:

$$\mathcal{S}^{(i)} = \{(x_j, \bar{y}_j^{(i)})\}_{j=1}^m \qquad \bar{y}_j^{(i)} = y_i + r_j$$

where $r_j$ are sampled with replacement from $\{r_j\}_{j=1}^m$.

In our framework, because $g$ must apply to $p'$, each $g^{(i)}$ is trained with propensity weights $w$. To account for cases where $p$ and $p'$ differ, the $g^{(i)}$ are trained not on sample sets of size $m$, but rather, of size $\tilde{m}(w)$, where $\tilde{m}(w)$ is the *effective sample size* [24] given by:

$$\tilde{m}(w) = \frac{\text{mean}(\{w_i\}_{i=1}^m)}{\text{var}(\{w_i\}_{i=1}^m)}, \qquad w_i = w(x_i) \;\; \forall i = 1, \ldots, m$$

### B.2 Quantile regression

Quatile regression is a learning framework for training models to predict the $\tau$-quantile of the conditional label distribution $p(y|x)$. Just as training with the squared loss is aimed at predicting the mean of $p(y|x)$, training with the absolute loss $|y - \hat{y}|$ is aimed at the median. Quantile regression generalizes the absolute loss by considering a 'tilted' variant with slopes $\tau - 1$ and $\tau$:

$$Q_\tau(y, \hat{y}) = \max\{(1 - \tau)(y - \hat{y}), \tau(y - \hat{y})\}$$

## Appendix C  Experimental details

### C.1 Experiment 1: Quadratic curves

Here we set $f^*(x) = -0.8x^2 + 0.5x + 0.1$. $F$ and $G$ include quadratic functions, and $H$ to include linear functions. For uncertainty estimation we used vanilla bootstrap, and for propensity scores we used logistic regression. For lookahead, we set $\lambda = 4$, $\tau = 0.95$, use $k = 10$ bootstrapped models, and train for $T = 5$ rounds. The data includes $m = 25$ samples $x$ drawn from $N(-0.8, 0.5)$, and $y = f^*(x) + \epsilon$ where $\epsilon \sim N(0, 0.25)$. We use a $75 : 25$ train-test split. The three conditions vary only in $\eta$ with values $\eta = 0.75, 1.25$, and $3.5$.

Quantitative results are given in the table below:

|  |  | RMSE | Imp. rate | Imp. mag. |
|---|---|---|---|---|
| $\eta = 0.75$ | baseline | 0.349 | 0.857 | 1.109 |
|  | lookahead | 0.351 | 0.857 | 1.108 |
| $\eta = 1.25$ | baseline | 0.342 | 0.143 | -0.261 |
|  | lookahead | 0.424 | 0.714 | 1.065 |
| $\eta = 3.5$ | baseline | 0.342 | 0 | -35.13 |
|  | lookahead | 0.675 | 0.571 | 0.604 |

### C.2 Experiment 2: Wine quality

The wine dataset includes $m = 178$ examples and $d = 13$ features. We learn a quadratic $f^*(x) = \sum_i \theta_i x_i + \sum_i \theta'_i x_i^2$. $F$, $G$, and $H$ include linear functions. For uncertainty estimation we used residuals bootstrap, and for propensity scores we used logistic regression. For lookahead, we set $\tau = 0.95$, use $k = 20$ bootstrapped models, and train for $T = 10$ rounds. For $f$, we use SGD with a learning rate of 0.1 and 1000 epochs for initialization and 100 additional epochs per round. For $g$, each sub-model was trained with SGD using a learning rate of 0.1 and for 500 epochs. We set $\eta = 0.5$ and $\eta = 2$ for the fully and partially mutable settings, respectively.

### C.3 Experiment 3: Diabetes

The diabetes dataset includes $m = 442$ examples and $d = 10$ features. We set $f^*(x)$ to be a generalized additive model (GAM) with splines of degree 10 trained on the entire dataset and tuned

Figure 5: Sensitivity analysis of learning with misspecified $\eta$. (Left) Improvement rate of misspecified models, trained on a single $\eta$ and evaluated on data generated with varying values of $\eta$. (Right) The ratio between the improvement rate of misspecified and correctly-specified models (i.e., trained on the same $\eta$ on which they are evaluated).

using cross-validation. In the first setting, $F$, $G$, and $H$ include linear functions. In the second setting, $F$, $G$ are quadratic functions (i.e., linear in $x_i$ and in $x_i^2$) and $H$ remains linear. For uncertainty estimation we used quantile regression, and for propensity scores we used logistic regression. For lookahead, we set $\tau = 0.8$ and train for $T = 10$ rounds. For $f$, we use SGD with a learning rate of 0.05 and 1000 epochs for initialization and 100 additional epochs per round. For $g$, we use SGD with a learning rate of 0.05 and for 500 epochs. For both linear and non-linear settings we set $\eta = 5$, and normalize $y$ to be in $[0, 1]$.

## C.4 Sensitivity analysis

The experiments in the paper assume models are trained with the same $\eta$ used in evaluation. Here we evaluate the sensitivity of our method to the misspecification of $\eta$. We use the diabetes experimental setup, train four different models with $\eta \in \{1, 2, 5, 10\}$, and evaluate each on action outcomes generated with $\eta' \in \{0, \ldots, 30\}$. Figure 5 (left) shows the improvement rate of each model evaluated on varying test-time $\eta'$. As can be seen, improvement rates across $\eta'$ show an inverse-U patter. In most regimes performance is robust, although for large deviations between $\eta$ and $\eta'$ improvement rates deteriorate. To investigate this, in Figure 5 (right) we compare the improvement rate of the misspecified model (i.e., trained on a fixed $\eta$) to that of a correctly-specified model, and report the ratio.[8] The correctly-specified model serves as a benchmark on performance, and results show that misspecified models are competitive with this benchmark (except for extremely large values of $\eta'$).

## Footnotes

[8]Due to randomness in experimentation, the ratio can be larger than one.