[Reviews · NeurIPS 2020]

Review 1

Summary and Contributions: The paper proposes an interesting method named lookahead regularization aimed to optimize both accurate predictions and good actions/decisions derived from these predictions. The main contribution is therefore the possibility of combining predictions with the effects of human decisions derived from these predictions. The approach is represented as a learning objective that considers a lookahead regularization term, which makes use of an uncertainty model that is also learned. This helps to penalize models for which decisions do not improve the outcome at a sufficiently large rate \tau, where improvement is defined in a positive direction. The uncertainty model is represented as a problem of learning under covariate shift, using importance weighting through an approximate model based on logistic regression. The training method alternates the optimization of the desired function f, the uncertainty model g and the logistic regression model h. The proposed method relies on some assumptions: 1) users make decisions in the direction of the function f learned by the model, 2) those decisions have an impact on the distribution of the observed features, which can also be observed and, 3) the condition distribution on outcomes, p(y | x’) is invariant for any arbitrary marginal distribution p(x’) – no unobserved cofounders are considered, similarly to some of the previous literature. The authors experiment with both synthetic and real-world datasets. The reported results suggest that the proposed method has the potential to get closer to a ground-truth model as decisions are made using the proposed approach, in the context of the data considered in these experiments.

Strengths: I would like to remark as strengths of the paper: - The paper tackles an important problem for the use of machine learning models in decision making, in which the input features of the model are not static but may change according to actions induced by the model. This is highly relevant for the machine learning community interested in applications of models for decision making. The novelty relies on the combination of prediction and actions under the same learning objective. - The empirical evaluation is sound in the sense that the method seems to follow the ground truth direction as actions are executed based on the model and the results of these actions are considered through the look-ahead regularization term.

Weaknesses: The main weaknesses I see in this paper are: - Although technically sound and interesting, the assumption on p(y | x’) not changing is a very strong assumption that seems to contradict the purpose of the framework which is to encourage better decisions. As these decisions are better and better, the target y’ will change in a positive manner, and therefore likely p(y | x’) will change. Not considering may pose the risk of making the problem of feedback loops even worse, as the effect of p(x) that changes to p’(x) is being considered, but not p(y | x’). In the context of predictive policing for example, the actions may imply more arrests for particular neighborhoods (Ensign et al., 2018), i.e. p’(x), while p(y | x’) remains fixed but may be actually changing because of these actions. I think more concrete effect of this assumption should be discussed more thoroughly on the paper, by quantifying, theoretically or experimentally, the effect of these assumptions. - In the experiments, although the experiments are tested with different values of the step-size parameter in the real-world datasets, it is not explained how the choice of this parameter can be made in practice. This is important for the application of this method to other datasets/problems. - There is no mention to the computational complexity of the proposed method. What is the cost of the extra-regularization term and the alternating mechanism proposed to solve this problem? Ensign, D., Friedler, S. A., Neville, S., Scheidegger, C., & Venkatasubramanian, S. (2018, January). Runaway feedback loops in predictive policing. In Conference on Fairness, Accountability and Transparency (pp. 160-171).

Correctness: The method sounds correctly formulated under the stated assumptions. The empirical methodology considers both synthetic and real-world datasets. It would be interesting to see an actual example, e.g. a case study, that shows the effectiveness of the method for encouraging better decisions as it is claimed at the beginning of the paper.

Clarity: The paper is very well-written, easy to follow and has no major errors or typos.

Relation to Prior Work: I am not fully familiar with the literature in strategic classification and causal modelling, therefore I could not tell it is complete with certainty. However, the description of the literature seems sound and connects naturally with the aim of the paper.

Reproducibility: Yes

Additional Feedback: I would suggest to: - Quantify and discuss more thoroughly the effect of the assumption on the invariance of p(y | x’), which is very relevant in a range of application areas. - Clarify ways in which the step-size parameter could be chosen in practice. - Provide information on the computational complexity of the proposed objective, and the alternating solution proposed. - There could be some theoretical analyses that could be useful for accompanying the paper – for example in terms of upper-bounds of actions or similar. ----- Response to rebuttal ----- I am happy with the authors’ explanations, and would suggest the authors to add their ideas on assumptions around the covariate shift as future work and the computational complexity somewhere in the paper.


Review 2

Summary and Contributions: the authors propose a new paradigm for learning models that are not only good at the prediction task, but also useful in telling users what features to change to improve favorable outcomes

Strengths: this is a great problem setup, relevant to a lot of real-world contexts where it is important to have recourse,with a clean formulation, and a well-written paper with nice figures

Weaknesses: the weaknesses are few and far-between. I think the choice of evaluation metrics could have been discussed a bit more. Also, the choice between uncertainty quantification methods appraised a bit more. I'm not so impressed by the choice of datasets, but you have to start somewhere.

Correctness: everything appears to be correct

Clarity: yes

Relation to Prior Work: yes, the paper explains how it relates to past work, and what is new and different. I think all the important references are captured and explained. one newer reference that the authors may want to discuss: https://arxiv.org/abs/2002.06673

Reproducibility: Yes

Additional Feedback: generally good work


Review 3

Summary and Contributions: This paper presents lookahead regularization, which essentially boils down to the inclusion of a regularization term in the objective, in the context of a framework that casts the problem of balancing prediction and decision quality as one of model selection. The regularization term includes a separate uncertainty model that estimates the uncertainty in the predictive model for out-of-distribution data points.

Strengths: The paper is well-presented. The masking framework is an interesting approach that I haven't seen before. The methods appear solid, and the problem the authors try to solve is an important one. The paper seems relevant to the broader NeurIPS community.

Weaknesses: Assumption 2 is quite strong. I know you can't verify this assumption in practice, but it would at least be interesting to see how sensitive the gains presented are to violations of this assumption. Otherwise it seems like this method would be quite fragile. If not in this version, this should be addressed in a follow-up. I think the example in the second paragraph of the Introduction is not as compelling as I think it could be. Perhaps an example such as that of the unintuitive effect of asthma in pneumonia risk prediction, as in Caruana '15. Moreover, the proposed methodology bears many similarities to Hardt's recent work on performative prediction and strategic classification. In light of those works, I'm not sure if this paper represents a novel leap forward—particularly insofar as the proposed methodology rely on strong assumptions.

Correctness: The claims and method appear to be correct. The experiments are well-motivated, although I am not sure about the connection to real-world settings. There might be some concern about shift under actions.

Clarity: The labels for the panels in Figure 1 need to be corrected. While attractively presented, the figures have text that I think is a bit too small. There should be an 'a' inserted in front of 'masking operator' on line 112. Otherwise, there seem to be no major typos or grammatical errors.

Relation to Prior Work: The literature review seems sufficient, but it omits Hardt's very recent work on performative prediction (https://arxiv.org/pdf/2002.06673.pdf). I think there is a lot of overlap between these papers methods-wise. I also wonder if there is some connection to online learning that may have been missed, although I am not an expert in that particular field.

Reproducibility: Yes

Additional Feedback:


Review 4

Summary and Contributions: This paper introduces a framework to learn an accurate predictive model that promotes good actions. The idea is that the result of making decision based on a predictive model can lead to a data distribution shift. The authors propose a framework to learn an accurate predicative model that leads to data distribution shift with better outcome in the future, by regularizing the objective function with a look-ahead regularization, based on the uncertainty of the prediction after a decision made by the user. The new regularization encourages actions that improve outcomes. This setup is quite new to me but it seems to be an important problem in certain applications such as medicine. The proposed approach requires explicit model of how users use the model to take action and the authors assume that decision made by the users follows a gradient ascent type of algorithm based on the predictive function (which we aim to learn) to change the mutable part of the features in order to improve action. They further make a strong assumption that the relationship between the features and outcome variables is invariant as the decisions are made and feature distribution changes. This is too simplistic as mentioned by the authors. In order to optimize the objective function, the authors need to learn three elements: the predictive model that maps features vector to output, the uncertainty model the measure the uncertainty of the model prediction as a result of data attribution shift, and propensity function to use the predictive model for the shifted data in test. The authors assume specific parametric forms for each one of these functions and perform an alternating optimization process to learn them. In their empirical study, the authors assume they have access to the true predictive function f* and build the evaluation based on this. This is too unrealistic and not sure how much the result is biased. After rebuttal: Thanks to authors responses, I increased my score.

Strengths: Addresses a very important and difficult problem Novel problem and solution (to the best of my knowledge) Difficult empirical evaluation

Weaknesses: Strong simplifying assumptions Not convincing empirical evaluation

Correctness: The claims and method seem correct. I am not sure about the set up in empirical study where they needed access to the ground truth model f* and had to define it. There are also other aspects of the framework that changes from one data set to another. I recommend defining the set up of the experiments first and then using consistent set up for all data. As I said, the empirical study for this problem is not easy.

Clarity: It is well-written but it is difficult to comprehend the details.

Relation to Prior Work: I am not familiar with the related work so cannot comment on whether the related work section is comprehensive.

Reproducibility: No

Additional Feedback:

[Author Response · NeurIPS 2020]

We thank the reviewers for their constructive and thoughtful comments. We begin with addressing concerns raised by
several reviewers and then proceed with individual responses to each reviewer.

**Relation to Perdomo et al.'s recent paper on performative prediction.** Our paper differs from Perdomo et al.
(referenced as [31]) in several important ways. [31] focuses on a setting where a predictive model at time $t$ affects the
input distribution at time $t + 1$, a special case of which is users changing their features according to the model. For
this setting, they give conditions for when a retraining procedure reaches an equilibrium. Their focus is exclusively on
minimizing predictive loss (through empirical risk minimization), and the predictive model matters through its effect on
convergence. They do not consider the quality of decision outcomes, and outcomes may be arbitrarily bad both in the
equilibrium and throughout the retraining process.

Whereas Perdomo et al. care only about predictive accuracy, we care also about the quality of the actual outcomes
associated with decisions, and study the tradeoff between decision improvement and predictive accuracy. This is a
substantive difference that entails the need for assumptions that provide causal validity (see below). Another important
difference is that we consider a one-time interaction between users and the model. Consider, for example, mortgage
buyers, ICU patients, or first-time medical consultation (e.g., oncology, cardiology, psychiatry, screening tests). Our
model is also relevant in settings with feedback coming in slowly, with models being intermittently re-trained, and
where decision outcomes are consequential at each step of the retraining process. The main technical contribution is in
extending the conventional risk minimization paradigm to account for decision outcomes.

**Assuming covariate shift.** Our work considers how decisions affect actual outcomes. This is a causal problem, and
as such, requires assumptions that ensure causal validity (in our work this comes in through our use of propensity
scores for reweighting evidence). The assumption of covariate shift (or its analogs, e.g., [34]) is made elsewhere in the
machine learning literature [32,29,33,38,41,45,12], and although this is a strong assumption, it is reasonable for many
applications. There exist important domains in which violations are sufficiently minor that this is a reasonable model,
see Mueller et al. [29] and Peters et al. [32] for discussion (giving writing improvement as one example domain).

As some reviewers suggest, it would be interesting to quantify the effects of relaxing covariate shift on our framework.
One possibility would be to assume Lipschitzness—in our case, that $p(y|x)$ changes smoothly with changes to $p'(x)$.
Allowing for changes in conditional outcomes would affect the correctness of propensity weights, but can be accounted
for by smoothly increasing uncertainty intervals, or equivalently, reducing the confidence $\tau$ in regard to improving
decisions. Quantifying the relation between the Lipschitz coefficient and $\tau$ looks interesting for future work.

Reviewer 1 suggests that the covariate shift assumption contradicts the purpose of the framework, raising questions
about feedback loops. We respectfully disagree. Undesired outcomes, as in the policing example, are the result of
sample bias in data (missing observations) and of the inappropriate use of predictive tools for decision making. Our
regularized predictor doesn't remove the problem with missing data, but nor is it adding to the problem; on the contrary,
through this framework one can avoid decisions that are not supported by data. In policing, covariate shift is a strong
requirement, but can be motivated through a suitable choice of $x$ that includes relevant information (e.g., location, kind
of policing, kind of community messaging), and while decisions are likely to affect the marginal $p(y)$, we do not see a
clear reason for why this would change the conditionals $p(y|x)$.

**Other individual responses:**

**[R1] Choosing** $\eta$: If decisions $x'$ are observed (and $\eta$ is assumed to be independent of $f$), then fitting $\eta$ to a sample set
$\{(x_i, x'_i)\}_{i=1}^m$ reduces to a unidimensional search problem. Similar assumptions on the decision model (often in the
form of cost functions) are common in the literature on strategic classification as well as recourse.

**Computational complexity:** Let $c(\cdot)$ be the cost of a forward pass, then $c(R) = O(c(\nabla_f(x)) + c(\ell(x')))$. For
linear $f, g, h$ and $x \in \mathbb{R}^d$, we have $c(\nabla_f(x)) = d$ and $c(\ell(x')) = kd$ where for quantile regression $k = 1$ and for
bootstrapping $k$ is the number of bootstrapped models. The cost of the alternating procedure follows accordingly.

**[R2]** Thank you for your encouraging and positive review.

**[R3] Example in intro:** Thank you for this comment, we will revise accordingly.

**Relation to Perdomo et al. paper:** Please see above. Note that we cite it as [31].

**[R4] Empirical evaluation:** As you note, the counterfactual nature of our setting makes validation challenging. Many
works face this challenge, and lacking query access to arbitrary inputs, must resort (as we do) to some form of simulation.
We have been careful to follow best current practices (e.g., [39]). Note that our approach does not require access to (nor
the existence of) a true model $f^*$, and we simply use $f^*$ to generate counterfactuals $y' \sim p(y|x')$ needed for evaluation.

[Meta-Review · NeurIPS 2020]

summary: The authors consider the problem of learning a classifier in the setting where the classifier output influences the system in turn. They present an approach to trade-off classifier quality with “lookahead regularization”, i.e. potential beneficial impact of classifier prediction on system behavior. pros: - important aspect not often studied: impact of predictions on behavior - clearly written, instructive illustrations - relevant literature mostly discussed cons: - $\Gamma$ must be known: assumption that a causal model is available meta-review: To me, this is a borderline paper for the following reason: I am not convinced that the basic assumption (which underlies the whole paper), that the prediction method has to also serve to make intervention recommendations, is sensible. None of the introductory examples convinces me of this. Consider the wine example, where one task is to predict wine quality and the other is to improve wine quality with policy recommendations -- I do not see the necessity to conflate them, I think in practice it perfectly valid to output both: a quality prediction tool (potentially taking into account spurious correlations), as well as a causal analysis of what should be changed to improve quality. The latter is actually assumed known in the paper in the form of the masking $\Gamma$. In general, I think this paper could contribute in a non-constructive way to the already murky discussion on the relation of predictions and interventions. As all reviewers favourably assessed the paper positively and it is not technically flawed, it is accepted. However, I urge the reviewers to reflect and discuss in the paper the basic assumption that a single classifier should take on two different tasks of prediction and policy recommendation; just because this is current practice in some applied ML system does not mean that this is desirable. The paper could be made much stronger IMO if it could convince a reader like me, to whom this setting looks like “narrow framing” for the sake of technical tractability, without considering its justification nor plausible alternatives.